# BMAS: A Brain-Inspired Multi-Agent System with PFC-Guided Task Coordination and Hippocampus-Neocortex Dual Memory for Scalable Multi-Step Reasoning

## Abstract

Multi-Agent Systems (MAS) leveraging large language models have shown promise in tackling complex, multi-step tasks by distributing responsibilities across specialized agents. Recent research has introduced mechanisms such as explicit role assignment, hierarchical task allocation, and dynamic coordination strategies to improve agent collaboration and execution quality. However, most efforts focus separately on task scheduling or information sharing, overlooking the interplay between system-level architecture and memory management. This gap gives rise to two coupled challenges: static architectures limit adaptive coordination, which in turn amplifies the accumulation of long interaction histories, leading to inefficiency, instability, and degraded scalability. Inspired by the human brain, we propose the Brain-inspired Multi-Agent System (BMAS), a principled framework that explicitly links architectural design with memory management. BMAS introduces a prefrontal cortex (PFC) like module responsible for hierarchical task decomposition, dynamic coordination, and working memory management, addressing the rigidity of conventional MAS architectures. Complementing this, a hippocampus-neocortex-inspired dual-memory system enables selective consolidation and semantic retrieval of task-relevant information, mitigating context inflation and enhancing backtracking capabilities. We validate BMAS on diverse complex task datasets in mathematics and coding, demonstrating superior performance in accuracy, efficiency, and reasoning stability compared to existing MAS. Our approach bridges neuroscience insights and MAS design, providing a scalable framework for collaborative AI systems requiring long-horizon reasoning. The code is available at https://anonymous.4open.science/r/BMAS-AAD0.

## 1 Introduction

Multi-agent systems (MAS) have emerged as a promising paradigm for tackling complex, multi-stage tasks that exceed the capability of individual agents (Hong et al., 2023; Wu et al., 2023a). By distributing responsibilities across specialized agents and enabling coordinated decision-making, MAS offer enhanced scalability, robustness, and adaptability (Chen et al., 2023; Liu et al., 2023). These properties make MAS particularly suitable for real-world scenarios requiring collaborative problem-solving and flexible reasoning.

To realize these benefits, recent research has introduced mechanisms such as explicit role assignment (Yue et al., 2025; Hong et al., 2023), hierarchical task allocation (Fosong et al., 2023; Kannan et al., 2024), and dynamic coordination strategies (Liu et al., 2023; Chen et al., 2023). While these methods improve controllability and execution quality by regulating agent collaboration, most efforts treat task scheduling and information sharing in isolation, overlooking the interplay between architectural design and memory management. As a result, two critical challenges remain unresolved in current MAS development.

The first challenge lies in **architectural limitations**. The overall design of MAS critically shapes efficiency and stability, yet agents are often predefined and connections fixed, preventing adaptive

reconfiguration in response to changing tasks or environments (Cemri et al., 2025). Importantly, an ill-structured architecture not only causes bottlenecks in information flow and decision-making but also exacerbates the problem of memory. Without principled planning of coordination pathways, information can be redundantly stored, inconsistently shared, or even repeatedly circulated among agents, leading to bloated and disordered memory states. Such entanglement between poor structure and chaotic memory ultimately destabilizes the entire system. Existing studies on modular and hierarchical architectures (Hong et al., 2023; Wang et al., 2025) and role allocation and task scheduling (Zhang et al., 2024b) highlight the importance of clearer structures, yet most designs remain ad hoc and fail to integrate systematic memory management, thus limiting robustness and generalizability (Wang et al., 2024; Li et al., 2024).

The second challenge concerns **memory and long-context management**. As task complexity increases, particularly in scenarios requiring long-term sequential reasoning, the accumulation of historical information becomes a major bottleneck. In MAS, repeated inter-agent communication amplifies this problem, creating scalability and responsiveness concerns (Li et al., 2023; Yan et al., 2025). Crucially, insufficient or poorly organized memory does not only degrade retrieval efficiency but also constrains structural flexibility. Without reliable memory to guide what should be preserved, consolidated, or recalled, agents are forced to operate on rigid routines or redundant exchanges, causing the system architecture to become stagnant and inefficient. Prior studies have explored external memory mechanisms (Abu-Rasheed et al., 2024; Liu et al., 2024) and memory compression techniques (Ge et al., 2023), as well as phase-based summarization (Zhou et al., 2022; Teng et al., 2025). Yet these approaches remain limited: external memory may trigger disordered retrieval and repetitive loops (Cemri et al., 2025), while phase-based methods risk summarization bias (Ramprasad et al., 2024) and lack mechanisms for reflection or backtracking. Without a co-designed architecture to anchor them, such memory solutions often fail to improve overall system adaptivity.

Together, these two challenges underscore the lack of **a unified and systematic framework** for MAS design. Without an architecture that jointly accounts for structural adaptability and efficient memory management, MAS remain vulnerable to instability, redundancy, and scalability bottlenecks when tackling complex tasks.

Although these challenges persist in MAS development, it is noteworthy that the human brain has naturally evolved highly efficient mechanisms for orchestrating complex, multi-step problem solving. Specifically, the brain assigns distinct functions to specialized cortical regions and achieves effective integration and utilization of historical information through a task-oriented dual-memory system that supports adaptive task simplification (Rusu & Pennartz, 2020; Unterrainer & Owen, 2006; Goel & Grafman, 1995; Barraclough et al., 2004).

The first mechanism involves **hierarchical task management**, where distinct cortical regions coordinate closely to decompose complex tasks into manageable subtasks and iteratively simplify them based on feedback. The prefrontal cortex (PFC) acts as a central planner, executing subtasks sequentially and refining task structure after each step.

The second mechanism is **an integrated dual-memory system**. The PFC manages working memory to retain only task-relevant information (Funahashi, 2006), while historical information is consolidated into long-term memory under hippocampal control (Ericsson & Kintsch, 1995; Kryukov, 2008; Winocur et al., 2010). When needed, the PFC can selectively retrieve long-term memories based on semantic cues (Funahashi & Andreau, 2013), avoiding interference from redundant or irrelevant information.

Inspired by these mechanisms, we propose a **Brain-inspired Multi-Agent System (BMAS)**. BMAS integrates principles from specialized brain regions to guide architectural design and incorporates **hierarchical task simplification** together with a task-oriented **dual-memory system** to address the challenges of static agent connections and long-context accumulation in MAS. Specifically, BMAS introduces a **PFC-like module** as the task planning and coordination hub, responsible for hierarchical decomposition and dynamic adjustment of tasks based on feedback. This module maintains a working memory for relevant contextual information and interacts with a **Hippocampus-like Memory Module** to enable selective storage and semantic retrieval of inactive task information.

Our contributions are summarized as follows:

- We propose a Brain-inspired Multi-Agent System (BMAS), drawing inspiration from the collaboration mechanisms between the PFC and hippocampus in the human brain. BMAS addresses the rigidity of static MAS architectures through hierarchical task simplification and dynamic coordination among agents.
- We design a task-oriented dual-memory system that models the complementary roles of the hippocampus and neocortex. This system enables selective consolidation and semantic retrieval of historical task information, effectively mitigating context inflation in long-horizon reasoning and improving system stability and backtracking capability.
- We validate the effectiveness of BMAS on three complex task datasets in mathematics and coding, demonstrating improved task execution efficiency, reasoning robustness, and adaptive memory utilization.

## 2  RELATED WORK

This section reviews existing research relevant to our work, focusing on two primary areas: architectural design in MAS and memory mechanisms for complex tasks.

### 2.1  MAS ARCHITECTURE DESIGN

Existing MAS mainly adopt two architectural paradigms: modular & hierarchical structures, and dynamic task scheduling. For example, MetaGPT (Hong et al., 2023) organizes agents in a "corporate" structure with roles such as product manager, developer, and tester, coordinating tasks via predefined workflows. While effective in structured domains, this design struggles with open-ended tasks. To increase flexibility, Dynamic LLM-Agent Network (Liu et al., 2023) enables agents to adjust collaboration dynamically, though it still depends on a central scheduler for global decisions, which incurs high resource use and synchronization issues. Another representative system, AgentVerse (Chen et al., 2023), reconfigures teams based on environment states and incorporates reflection after each round; however, frequent regeneration of agent descriptions and synchronization of contexts introduce considerable computational overhead.

### 2.2  MEMORY MECHANISMS FOR COMPLEX TASKS

To reduce redundancy and improve reasoning efficiency in long-chain tasks, recent studies propose various memory management strategies. One line of work segments tasks into phases and summarizes at each phase's end to reduce context size (He et al., 2024), but such approaches are vulnerable to summarization bias (Ramprasad et al., 2024) and offer limited support for reflection or backtracking. Other methods, such as Least-to-Most Prompting (Zhou et al., 2022), decompose problems into subproblems and summarize incrementally, yet they lack explicit mechanisms for backtracking or automatic retrieval. More recently, Atom of Thought (Teng et al., 2025) retains only current key facts to minimize information load, but this may cause early critical information to become inaccessible, allowing errors to propagate without explicit checking.

## 3  METHOD

### 3.1  OVERALL STRUCTURE

Drawing on the brain's specialized cortical regions and its dual memory system of hippocampus and neocortex, our BMAS framework integrates five core modules—Task Decomposition Agent, Instructor Agent, Actor Agent, Confidence Evaluator, and Rethink Agent—together with a two-tier memory architecture comprising Working Memory and Long-term Memory. See Figure 1 for an overview.

### 3.2  BRAIN-INSPIRED ARCHITECTURE DESIGN

Drawing on the brain's specialized regions and their interactions, BMAS maps each cortical function to a corresponding agent:

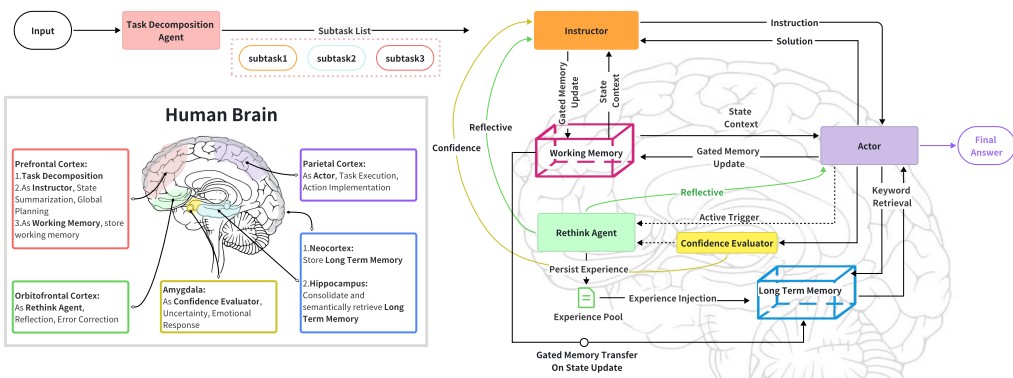

Figure 1: Overall Structure of BMAS: The left side illustrates the main functional regions of the human brain, while the right side presents the proposed Brain-inspired Multi-Agent System (BMAS) framework. Functional correspondences between brain regions and system modules are indicated through color coding and spatial alignment.

- **Task Decomposition Agent ⇔ PFC:** Simulates the PFC's role in breaking down a complex task into clear, executable subtasks.
- **Instructor Agent ⇔ PFC:** Mirrors the PFC's decision-making: simplifying the current task state and issuing instructions to the Actor Agent.
- **Actor Agent ⇔ Parietal Cortex (PC):** Models the PC's execution of PFC instructions (Caminiti et al., 2017; Turella et al., 2020)—writing code, performing mathematical reasoning, invoking tools—and aggregates results into a solution to feed back to the Instructor.
- **Confidence Evaluator ⇔ Amygdala:** Emulates the amygdala's uncertainty detection by assessing solution confidence and informing the Instructor to adopt a more cautious strategy (Grupe & Nitschke, 2013; Kim et al., 2011).
- **Rethink Agent ⇔ Orbitofrontal Cortex (OFC):** Replicates the OFC's reflective and corrective function (Mansouri et al., 2014; Howard & Kahnt, 2018), evaluating solutions against the original goals and suggesting alternative approaches when confidence is low or errors are signaled by the Actor.

In operation, once a user submits a problem, the Task Decomposition Agent first segments it into subtasks. For each subtask, the Instructor Agent issues instructions that the Actor Agent carries out. The Actor Agent's Solution and its confidence score (from the Confidence Evaluator) may trigger the Rethink Agent to propose refinements. This loop continues until the final subtask is successfully completed, at which point the Actor Agent delivers the overall solution.

### 3.3 HISTORICAL INFORMATION MANAGE

Humans retain only task-relevant data in working memory while consolidating important experiences into long-term memory, minimizing cognitive load and preserving critical information. This dual-memory mechanism addresses the Historical Information Management Bottleneck in Section 1 Introduction.

Inspired by this, our memory system comprises two complementary modules:

- **Working memory:** Modeled on the PFC, it dynamically holds only information closely related to the current subtask.
- **Long-term Memory:** Modeled on neocortical storage under hippocampal control, it selectively consolidates filtered working-memory traces and supports semantic retrieval by both the Instructor and Actor agents.

In our framework, each subtask is represented by a structured system state that evolves as the subtask progresses. When the system transitions from one subtask state to the next, working memory contents are transferred to long-term memory and reset. Hippocampal-inspired inhibitory mechanisms filter

out irrelevant or failed information, encoding only salient instruction–solution pairs (Lisman & Grace, 2005), formalized by a heuristic filter function $\text{Filter}(\cdot)$.

PFC–Hippocampal feedback further enables new memories to integrate with existing ones, allowing related experiences to be recalled together (Born & Wilhelm, 2012). Let the working memory at dialogue round i contain n instruction–solution tuples $(\mathfrak{I}_k, \mathfrak{S}_k)$. We update the long-term memory $M^L$ as follows:

$$M_i^L = M_{i-1}^L \cup \{ \text{Filter}\left(\{ (\mathfrak{I}_k, \mathfrak{S}_k) \}\right) \mid k = 1, 2, \ldots, n \} \tag{1}$$

For retrieval, the Instructor issues a semantic query q to the hippocampal module (Raposo et al., 2009; Preston & Eichenbaum, 2013), supporting task-oriented long-term memory. It returns two subsets of long-term memory:

$$\text{R}(q) = \left( M_{\text{task}}^L(q),\ M_{\text{exp}}^L(q) \right) \tag{2}$$

where $M_{\text{task}}^L(q)$ contains task-specific memories filtered from past working memory, and $M_{\text{exp}}^L(q)$ provides analogous experiences from other tasks. This hierarchical retrieval enables agents to ground reasoning in both current task history and broader experiential knowledge, aligning with our framework's focus on hierarchical task simplification and task-oriented dual-memory.

### 3.4 TASK DECOMPOSITION AND DYNAMIC SIMPLIFICATION MECHANISM

Building on our brain-inspired architecture and task-oriented dual-memory framework, we develop a hierarchical decision-making process that integrates dynamic subtask management, Markovian instruction generation, and adaptive reasoning—closely mirroring human problem-solving.

**Task Decomposition:** The Task Decomposition Agent splits the problem into subtasks, activating the first one. The Instructor Agent monitors progress and advances to the next subtask upon completion, optionally invoking a task-optimizing state transition. After the final subtask, the Actor Agent outputs the overall solution.

**Dynamic Simplification:** Humans continually refine their strategy as they reason. To emulate this, our system dynamically simplifies the remaining problem before issuing new instructions. As the system progresses through each subtask, it updates an internal state representation that captures the current context and memory. We formalize the system state as $S = (I, M^W)$, where I is the state information (a concise description of the current subtask $\alpha_j$, the set of completed conditions C, the simplified problem statement P, global obstacles to avoid O, and the next promising direction N) and $M^W$ is the working memory. Storing both in a single state tuple reflects how the PFC maintains and integrates task context and short-term memory.

We model state updates as a Markov transition:

$$S_i = (I_i, M_i^W) \quad I_i \sim p(I \mid I_{i-1}, M_{i-1}^W) \quad M_i^W = \varnothing u \tag{3}$$

where p denotes the conditional probability distribution for state transitions. This formulation reflects the Markov property: the current state $I_i$ depends only on the immediately preceding state's information $I_{i-1}$ and working memory $M_{i-1}^W$.

**Markov Decision-Making Mechanism:** At each step, the Instructor (PFC-like) issues the next instruction $\mathfrak{I}_{k+1}$ conditioned on the current state $S_i$ the most recent solution $\mathfrak{S}_k$, its confidence $\mathfrak{C}_k$, and any reflection feedback $\mathfrak{R}_k$:

$$\mathfrak{I}_{k+1} \sim p\left( \mathfrak{I}_{k+1} \mid S_i, \mathfrak{S}_k, \mathfrak{C}_k, \mathfrak{R}_k^{\text{opt}} \right) \prod_{j=0}^{i-1} p(S_{j+1} \mid S_j) \tag{4}$$

Compared to full-history conditioning $p(a_i \mid a_{i-1}, \ldots, a_0)$, the Markovian transition mechanism generates each instruction based only on the current state $S_i$, simplifying the conditional space and

avoiding redundant modeling. Here, $S_i$ includes $I_i$, which summarizes relevant context and filters out irrelevant information. This approach reduces context inflation and noise in long-horizon tasks, and supports efficient reflection and confidence evaluation for timely error correction.

**Brain-Inspired Single-Threaded Adaptive Reasoning Mechanism:** Unlike parallel approaches such as "Tree of Thought" (Yao et al., 2023), humans reason along a single path, using self-correction and strategy shifts for creativity. In our system, the PFC-like Instructor reviews history to consider alternatives, while the OFC-like Rethink Agent provides feedback for adjustment. This adaptive, single-threaded process expands the solution space, maintains stable context, and minimizes information loss. Experiments on math and coding tasks show these mechanisms preserve critical context and correct errors effectively.

## 4 EXPERIMENT

We evaluate our framework on the American Invitational Mathematics Examination (AIME) and MATH (Hendrycks et al., 2021) mathematical benchmarks, as well as the HumanEval (Chen et al., 2021) code-generation dataset, and compare its performance against state-of-the-art (SOTA) MAS. All experimental configurations are provided in Appendix A of the Technical Appendices, which are included as a separate PDF in the paper's supplementary material.

### 4.1 PERFORMANCE ON AIME BENCHMARK

AIME is a prestigious U.S. high-school mathematics competition with problems covering algebra, geometry, number theory, and combinatorics. The AIME24 and AIME25 datasets each contain 30 problems from the 2024 and 2025 exams.

In our experiments, we focus on comparing two baselines:

- **SINGLE:** A single agent with external-tool invocation, engaging in up to 10 rounds of self-dialogue before producing a final answer. Empirically, 10 rounds yield optimal performance.
- **OWL:** An open-source multi-agent framework from the CAMEL (Li et al., 2023) team, based on "AI USER" and "AI ASSISTANT" roles (similar to our Instructor and Actor agents). OWL ranked first among open frameworks on the GAIA benchmark. Comparing BMAS with OWL highlights the benefits of our hierarchical task simplification and dual-memory architecture.

All three systems use Qwen3-Plus as the backbone model for its low cost, fast inference, and strong performance. Unless otherwise specified, temperature is set to 0.0 and deep-thinking modes are disabled. Each framework receives the same high-level task prompt and tool suite (e.g., search, code execution). Each problem is solved once. Full prompts and tool lists are provided in Appendix A.

Table 1: Accuracy of Qwen3-Plus on AIME dataset with different agent structure

|         | Base Model | Task Count | Agent Structure | Correct Count | Accuracy (%) |
| ------- | ---------- | ---------- | --------------- | ------------- | ------------ |
| AIME24  | Qwen3-Plus | 30         | SINGLE          | 5             | 16.67        |
|         |            |            | OWL             | 14            | 46.67        |
|         |            |            | **Ours**        | 18            | 60.00        |
| AIME25  | Qwen3-Plus | 30         | SINGLE          | 5             | 16.67        |
|         |            |            | OWL             | 13            | 43.33        |
|         |            |            | **Ours**        | 17            | 56.67        |

In Table 1, BMAS increases accuracy on AIME24 from 5/30 to 18/30, a 46.7% relative gain over SINGLE and a 16.7% absolute gain over OWL. On AIME25, BMAS scores 17/30, similarly outperforming SINGLE (5/30) and OWL (13/30) by 43.3% relative and 16.7% absolute margins.

### 4.2 PERFORMANCE ON MATH BENCHMARK

The MATH dataset covers geometry, algebra, probability, number theory, prealgebra, and precalculus, divided into five difficulty levels. Following Data Interpreter (Hong et al., 2024), we select four cate-

gories—probability, number theory, prealgebra, and precalculus—and focus on the most challenging level 5 problems (i.e., MATH$_{\text{Level 5}}$).

We compare our framework with MathChat (Wu et al., 2023b), AutoGen (Wu et al., 2023a), and Data Interpreter (Hong et al., 2024). MathChat uses a "student–teacher" dialogue paradigm, similar to our Instructor–Actor design. AutoGen allows manual construction of multi-agent teams with roles such as Questioner, Solver, and Verifier. Data Interpreter employs hierarchical graph modeling and dynamic planning, and is the current SOTA on the MATH benchmark. In contrast, our framework is general-purpose, requiring no task-specific role configuration or parameter tuning, thus offering greater adaptability and ease of use.

We conduct experiments in BMAS using both GPT-4 Turbo and GPT-4.1 Mini as the base models. Table 2 compares our framework's accuracy on different MATH$_{\text{Level 5}}$ categories with other leading

Table 2: Accuracy (%) on MATH$_{\text{Level 5}}$ with different agent structures. All results are based on GPT-4 Turbo, except for the last row, which uses GPT-4.1 Mini. "C.Probability" denotes the "Counting and Probability" category.

|  | C.Probability | Number Theory | Prealgebra | Precalculus |
|---|---|---|---|---|
| MathChat | 52 | 60 | 60 | 19 |
| AutoGen | 59 | 66 | 63 | 12 |
| Data Interpreter | 68 | 82 | 74 | 29 |
| **Ours (GPT-4 Turbo)** | 71 | 84 | 76 | 44 |
| **Ours (GPT-4.1 Mini)** | 75 | 96 | 90 | 82 |

multi-agent systems. Our framework achieves the largest improvement on Precalculus, with a 15% increase over the previous SOTA using GPT-4 Turbo, mainly due to our multi-stage reasoning, dynamic memory management, and reflection mechanisms.

For Number Theory, we reach 96% accuracy with GPT-4.1 Mini. In contrast, gains on Counting and Probability are smaller, as these tasks rely more on external knowledge and probabilistic modeling, while our framework is optimized for symbolic reasoning and structured decomposition.

To improve performance on probability problems, we increased the confidence threshold for the rethink agent and added prompts encouraging the use of code tools for case enumeration. After these adjustments, accuracy improved from 61% to 71%, demonstrating the effectiveness of our error correction mechanism and the importance of guiding agents to use appropriate tools.

### 4.3 PERFORMANCE ON CODING BENCHMARK

We evaluate BMAS on the HumanEval code-generation dataset (Chen et al., 2021), which contains 164 programming problems with function signatures, descriptions, reference solutions, and test cases. We report pass@1 accuracy using GPT-4.1 mini as the base model.

We compare BMAS with several state-of-the-art multi-agent frameworks for code tasks: MetaGPT (Hong et al., 2023) uses a company-like structure with fixed agent roles and workflows, suitable for standardized code generation. In contrast, BMAS adopts adaptive task decomposition and dynamic simplification. AgentVerse (Chen et al., 2023) dynamically adjusts expert teams and incorporates reflection after each round, but requires frequent regeneration of expert descriptions and different context synchronization compared to BMAS. MapCoder (Islam et al., 2024) assigns agents to specialized functions, with a Retrieval Agent focused on external knowledge bases, differing from BMAS's selective long-term memory retrieval. AFlow (Zhang et al., 2024a) decomposes tasks and manages execution dependencies, accumulating all intermediate results, while BMAS emphasizes selective retrieval and dynamic simplification.

Table 3: Accuracy on HumanEval dataset with different agent structure

|  | MetaGPT | AgentVerse | MapCoder | AFlow | Ours |
|---|---|---|---|---|---|
| Accuracy | 85.9 | 89.0 | 93.9 | 94.7 | 97.0 |

As shown in Table 3, our approach achieves the highest pass@1 accuracy among existing frameworks. This improvement is due to our dual-memory management, which controls context and prevents redundant code generation, as well as hierarchical task decomposition and multi-round dialogue, which ensure all specification constraints are addressed and enable cross-validation by multiple tools.

Our framework achieves significant gains on both mathematical and code tasks, closely linked to our brain-inspired design. Inspired by the brain's dynamic coordination of cortical regions, we introduce adaptive task decomposition and collaboration mechanisms. The Task Decomposition Agent adjusts its strategy based on task type, while the Actor and Instructor flexibly invoke appropriate tools—using symbolic computation tools for equations and code tools with automatic test case generation for code tasks. Experiments show this approach enhances generalization and adaptability.

## 4.4 ABLATION STUDY

In this ablation study, we focus on two key aspects: (1) evaluating the performance gains of the multi-agent framework over single-agent baselines and quantifying the impact of core mechanisms on overall accuracy; and (2) analyzing how the framework's accuracy evolves with the number of interaction rounds, thereby demonstrating its stability and advantage in complex, multi-step tasks.

Our core mechanisms include task decomposition and dynamic simplification, as well as a dual-memory module comprising long-term and working memory. Since these mechanisms operate in tandem, they are ablated jointly in our experiments. In the ablated system, only working memory is retained, which stores all information from the beginning to the end of the task—consistent with the memory strategies adopted by most existing multi-agent frameworks such as MetaGPT (Hong et al., 2023) and CAMEL (Li et al., 2023).

We randomly sampled 100 level-5 problems from the MATH benchmark (i.e., $MATH_{Level\,5}$) and additionally incorporated the AIME24 and AIME25 datasets, which present greater challenges than MATH. These two types of tasks enable a comprehensive evaluation of our framework's performance across varying levels of problem difficulty.

**Impact of Removing Core Mechanisms:** We compare (1) CoT (Wei et al., 2022), where a single agent generates answers; (2) an ablated multi-agent framework without our task management and memory mechanisms; and (3) our full framework in Figure 2.

On the $MATH_{Level\,5}$ dataset, both our framework and the ablated version outperform CoT, as multi-agent settings enable tool use, reducing calculation errors. Our framework achieves slightly higher accuracy, as the ablated system often answers too quickly without sufficient verification.

On AIME24 and AIME25, the differences are more pronounced. CoT only solves the simplest problems, while the ablated framework, despite tool use, suffers from repeated tool calls and inefficient, trial-and-error reasoning due to lack of memory management. In contrast, our framework uses long-term memory and task summarization to avoid redundant work and enables effective strategy adjustment through reflection. This design allows multiple opportunities for error correction and prevents reverting to previous mistakes.

Overall, our mechanisms are especially important for complex, multi-step reasoning tasks like AIME, while on simpler $MATH_{Level\,5}$ tasks, accuracy gains come with increased resource use. To improve efficiency, we reduce unnecessary subtasks and allow the instructor to skip them. In the future, we plan to add adaptive mechanisms for dynamic switching between fast and slow thinking modes (Su et al., 2024).

**Accuracy vs. Number of Interaction Rounds:** We study how task accuracy changes with the number of agent interaction rounds. More complex tasks generally require more rounds, and in conventional frameworks, longer dialogues often reduce accuracy.

We compare our full framework with the ablated version on $MATH_{Level\,5}$ and AIME datasets, analyzing the distribution of interaction rounds and accuracy at different thresholds. This shows how accuracy evolves with more reasoning steps.

As shown in Figure 3, on $MATH_{Level\,5}$, the ablated framework usually finishes in 1–3 rounds (accuracy $\sim$0.85), while our framework typically takes 8–11 rounds (accuracy $\sim$0.95), achieving a higher average accuracy (0.90 vs. 0.85). On AIME, the ablated version operates in 3–5 rounds,

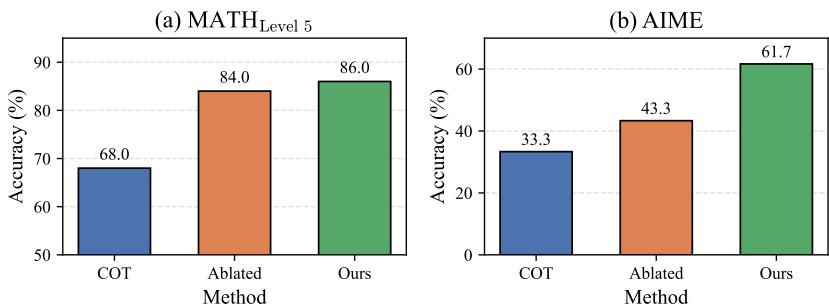

Figure 2: Comparison of Accuracy among CoT, Ablated Framework and Our Framework on MATH$_{Level\,5}$ (a) and AIME (b) Datasets

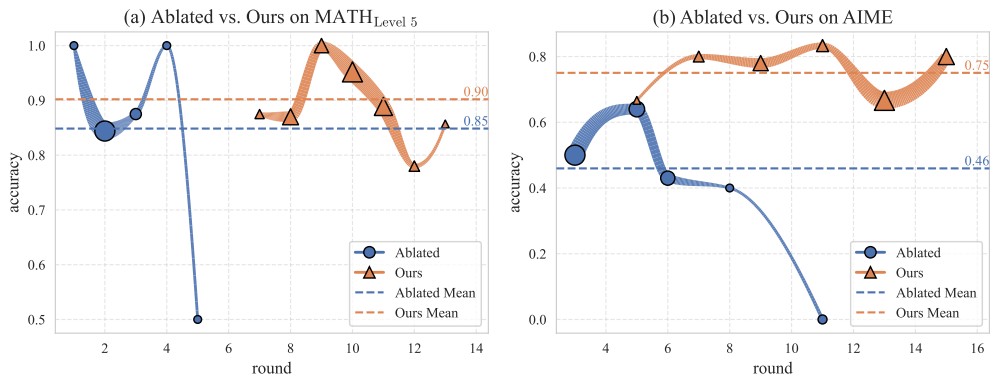

Figure 3: Comparison of ablated and original frameworks on the relationship between dialogue rounds and accuracy for (a) MATH$_{Level\,5}$ and (b) AIME tasks

but its accuracy drops after round 6 due to context length, while our framework maintains stable performance in 13–15 rounds by summarizing and transferring information to long-term memory. This demonstrates the effectiveness of our long-term memory and dynamic task simplification in preserving accuracy during multi-step reasoning.

## 5 CONCLUSION

We present BMAS, a brain-inspired multi-agent framework that significantly enhances the ability of LLM-based agents to solve complex, multi-step reasoning tasks in mathematics and coding. Drawing inspiration from coordinated interactions among cortical regions, BMAS integrates a PFC-like module for hierarchical task decomposition and dynamic coordination with a hippocampus–neocortex-inspired dual-memory system for task-oriented memory management and selective long-term retrieval. By incorporating an amygdala- and OFC-driven reflection loop, the system actively detects and corrects errors, adaptively refining its reasoning strategy. This architecture enables agents to decompose and simplify complex problems efficiently, maintain coherent information flow, and leverage past experiences to improve reasoning performance. BMAS demonstrates superior performance on tasks requiring long reasoning chains, while for simpler tasks it matches the accuracy of single-agent baselines, albeit with increased computational overhead. Future work will focus on enhancing BMAS's task scheduling flexibility and adaptive decomposition strategies within complex mathematics and coding tasks, enabling the system to dynamically adjust subtask allocation and execution order based on task complexity and feedback.

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

## A  APPENDIX

## B  EXPERIMENTAL SETUP

### B.1  EVALUATION METRIC

All experiments are evaluated using the pass@1 metric. Formally, pass@1 is defined as follows:

$$\text{pass@1} = \frac{1}{N} \sum_{i=1}^{N} \mathbb{I}(A_i) \tag{5}$$

where $N$ denotes the total number of test samples, and $\mathbb{I}(A_i)$ is the indicator function. The value of pass@1 ranges from 0 to 1, with higher values indicating a stronger ability of the model to generate correct answers on the first attempt.

## B.2 PROMPT TEMPLATES

We use the same task prompt templates for all agent frameworks. These prompts are adapted from the baseline prompt templates provided in the GitHub repository (`https://github.com/qixucen/atom`). The templates incorporate basic chain-of-thought (CoT) guidance to ensure prompt quality, while remaining simple and avoiding additional optimizations.

**Prompt for mathematical tasks:**

> You are a precise math problem solver. Solve the given math problem step by step.
>
> QUESTION: `{question}`
>
> Please extend your chain of thought as much as possible; the longer the chain of thought, the better.
>
> You can freely reason in your response, but please enclose the final answer within `<answer></answer>` tags (pure number without units and explanations)

**Prompt for coding tasks:**

> You are a precise coding problem solver. Solve the given coding problem step by step.
>
> QUESTION: `{question}`
>
> Please extend your chain of thought as much as possible; the longer the chain of thought, the better.
>
> You can freely reason in your response, but please enclose the final answer within `<answer></answer>` tags (Only code, no explanations. Comments are optional)

## B.3 AVAILABLE TOOLSET FOR AGENTS

**Basic Toolset:**

> **SearchToolkit**: Provides access to various search engines, including Tavily, Linkup, Google, DuckDuckGo, Brave, and Wikipedia, enabling agents to retrieve relevant information from the web.
>
> **MathToolkit**: Offers five fundamental mathematical operations: addition, subtraction, multiplication, division, and rounding.
>
> **SympyToolkit**: Integrates a wide range of symbolic computation and linear algebra functionalities, including expression simplification, expansion, factorization, polynomial representation, degree and coefficient extraction, as well as solving equations and inequalities (linear, nonlinear, univariate, and multivariate), and root finding. All functions accept input in the form of strings or lists and return structured JSON results.
>
> **CodeExecutionToolkit**: Executes given Python code snippets within a sandboxed environment to ensure security and isolation.
>
> **ThinkingToolkit**: Facilitates the recording and management of reasoning chains, covering the entire process from planning, hypothesizing, thinking, contemplating, critiquing, synthesizing,

reflecting, to brainstorming ideas. Users can sequentially document each step, forming an organized chain of thought, and can recall recent reasoning steps (`recall_recent_chain`) or previously generated ideas (`recall_generated_ideas`) at any time.

**Additional Tools Introduced in This Work:**

The following tools are exclusively available to agents within our BMAS project; agents in baseline or comparison frameworks do not have access to these functionalities.

**LongTermMemoryToolkit**: Retrieves relevant long-term memories from a vector database based on semantic similarity to the provided keywords, and also searches an experience pool for related experiences. Both sources of information are returned to the agent.

**GenerateToolkit**: Automatically generates multiple diverse solution strategies for a given task using a large language model, suitable for exploratory and creative tasks. This tool does not directly provide final answers for mathematical or coding problems, but instead outputs detailed solution approaches. Inspired by the Tree of Thought framework, but unlike Tree of Thought, which requires manually constructed prompts for the generate operation (thus limiting automation and effectiveness), our approach enables the language model to autonomously perform the generation process.

**KeepBestNToolkit**: Utilizes a large language model to evaluate multiple candidate solutions based on the initial task description, assisting the agent in filtering out incorrect answers. Similar to the GenerateToolkit, this tool is inspired by the Tree of Thought framework.

## B.4 AIME EXPERIMENTS

**Datasets:** We obtained the 2024 dataset from Maxwell-Jia/AIME_2024 and the 2025 dataset from math-ai/aime25 on Hugging Face.

**Base Model:** For all three experimental groups, we use `qwen-plus-2025-04-28` as the base model, with the deep thinking mode disabled. Unless otherwise specified, the temperature is set to 0.

**Framework 1: Single-Agent**

*Configuration:* Up to 10 rounds are allowed. The input is the problem statement, and in each round, the model receives the user question along with all historical context as input. The model can autonomously invoke basic tools (such as reasoning, code execution, search, etc.), and the results of tool execution are automatically appended to the historical context. If the model's output contains the keyword "answer", the process terminates immediately and returns the answer; otherwise, the next round proceeds until the 10-round limit is reached.

**Framework 2: OWL Framework**

*Configuration:* We use the camel repository (version #2138) and the owl repository (version #332). Upon receiving the input question, the system initializes two agents: a user agent and an assistant agent. The reasoning process is conducted in a turn-based dialogue manner, where in each round, the assistant agent first generates a response based on the current context, followed by the user agent generating new instructions or feedback according to the assistant's reply, thereby progressively refining and advancing the task. Both agents have access to the entire historical context. The assistant agent is capable of invoking basic tools. The process is limited to a maximum of 20 rounds.

**Framework 3: BMAS (Ours)**

*Configuration:* Up to 20 rounds are allowed. The system consists of an Instructor Agent, Actor Agent, Task Decomposition Agent, Rethink Agent, and Confidence Evaluator. The temperature for the Rethink Agent is set to 0.4, and the confidence threshold for triggering reflection is 0.5. A dual-memory module is incorporated: in each round, the Instructor Agent and Actor Agent can only access the contents of the working memory as contextual information, but can actively retrieve relevant information from the long-term memory. The Actor Agent is able to utilize both the basic toolset and the additional tools introduced in our work(see Section B.3).

## B.5 MATH EXPERIMENTS

**Dataset:** We utilize all level 5 problems from the datafreak/Math-level-5 dataset on Hugging Face. Following the settings in previous works, we select four categories for evaluation: Counting & Probability, Number Theory, Prealgebra, and Precalculus. Other categories are primarily geometry-related, and since our framework is based on a unimodal large language model, it is more advantageous for non-geometry problems. Future work may extend our framework to multimodal models, enabling evaluation on geometry tasks.

**Base Model:** The results of other baselines are taken from published papers, and we have verified that all use GPT-4 Turbo as the base model.

**Our Experiments:** We evaluate our framework using both GPT-4 Turbo and the latest GPT-4.1 Mini as base models.

**Experimental Settings:** For most tasks, the maximum number of reasoning rounds is set to 20, with a confidence threshold of 0.5 for triggering the reflection mechanism. For Counting & Probability tasks, the maximum number of rounds is increased to 25, and the confidence threshold is set to 0.7.

**Analysis:**

(1) *Enhancing Accuracy via Advanced Error Correction Strategies:* Initially, the accuracy on Counting & Probability tasks was only 61%. Through the above adjustments and iterative optimization of system prompts, the accuracy increased to 74%. This improvement is attributed to the agent's tendency to perform direct reasoning, which often leads to incorrect results in such tasks. By explicitly instructing the agent to avoid direct reasoning and instead use code-based simulation to enumerate possible cases, and by increasing the confidence threshold to trigger more frequent reflection, the framework can better detect and correct erroneous reasoning. The Instructor Agent also monitors the Actor Agent for potential mistakes. During dynamic task simplification, the framework identifies the high risk of direct reasoning and proactively switches to more detailed simulation and enumeration strategies, demonstrating advanced levels of reasoning and self-adjustment.

This improvement in accuracy highlights the effectiveness of our reflection mechanism and the role of the Instructor Agent in guiding and correcting the reasoning process. (2) *Effect of Different Base Models:* Experimental results show that as the underlying language model improves (e.g., GPT-4.1 Mini compared to GPT-4 Turbo), the performance of our framework also increases significantly. This demonstrates the generalizability of our framework and its ability to effectively leverage and amplify the reasoning and decision-making capabilities of advanced language models.

## B.6 HUMANEVAL EXPERIMENTS

**Base Model:** GPT-4.1 Mini

Except for replacing the input prompt with the "Prompt for coding tasks" (see Section B.2), no modifications were made to our framework. All other experimental settings for our framework are consistent with those described in Section B.4. Nevertheless, our framework achieved excellent results on the code task set, demonstrating its strong scalability and adaptability.

## B.7 ABLATION EXPERIMENTS

**Datasets and Base Models:**

- **AIME Tasks:** For the AIME experiments, we used all data from AIME24 and AIME25 as described in Section B.4.
  **Base model:** Qwen3 Plus. This selection enables the reuse of results from Section B.4.

- **MATH Tasks:** For the four categories of MATH tasks described in Section B.5, we randomly sampled 25 problems from each category, resulting in a total of 100 tasks.
  **Base model:** GPT-4.1 Mini. This choice allows for the reuse of experimental results from Section B.5.

**Framework 1: CoT**

*Configuration:* The prompt described in Section B.2 was used, and the base model was directly called without additional agent mechanisms.

*Analysis:* Compared to the single-model CoT approach, the multi-agent framework achieves superior performance, which validates its effectiveness. This performance gain can be primarily attributed to two key factors: the incorporation of a reflection mechanism, and the collaborative interaction between the Actor Agent and the Instructor Agent. Together, these components enhance the correctness of the reasoning process and contribute to higher overall accuracy.

**Framework 2: Ablated Framework**

*Configuration:* Compared to the full framework (Ours), the ablated version introduces the following modifications: (1) The dual-memory module is removed, allowing all agents to access the entire historical context; (2) The Task Decomposition Agent is omitted, so no explicit task decomposition is performed; (3) Periodic state summarization is not conducted.

**Framework 3: BMAS (Ours)**

*Configuration:* The configuration is consistent with the experimental settings for our framework described in Section B.4, with no additional modifications. This group serves as the control for comparison.

## C    FRAMEWORK IMPLEMENTATION AND CASE STUDY

Our BMAS framework integrates five core modules—Task Decomposition Agent, Instructor Agent, Actor Agent, Confidence Evaluator, and Rethink Agent—together with a two-tier memory architecture comprising Working Memory and Long-term Memory.

### C.1    CORE MODULES

**Task Decomposition Agent:**

> **Divide this task into subtasks:** {task}
> **<thoughts>**
> You can use the following thoughts to solve the problem:
>
> - For coding problems, try to find solutions with optimal time and space complexity
> - For coding problems, follow this approach: requirement understanding – solution design – complexity analysis – code implementation – testing – handling edge cases – code review
> - Try to solve mathematical problems through code or sympy whenever possible
> - How can I simplify the problem so that it is easier to solve?
> - How can I break down this problem into smaller, more manageable parts?
> - What are the key assumptions underlying this problem?
> - **Critical Thinking:** This style involves analyzing the problem from different perspectives, questioning assumptions, and evaluating the evidence or information available.
> - Let's make a step by step plan and implement it with good notion and explanation.
> - What kinds of solution typically are produced for this kind of problem specification?
> - Given the problem specification and the current best solution, have a guess about other possible solutions.
> - What are the potential obstacles or challenges that might arise in solving this problem?
> - You should adapt the thoughts to the current task.
> **</thoughts>**
>
> **# Format Requirements:**
> - Number each subtask (e.g., "1. ", "2. ")

- One subtask per line
- No additional text
- 3-5 subtasks total, each subtask should not have sub-tasks
- Each subtask should be:Independent, Moderate complexity, Clear completion criteria

**Notice:**

**Violet** is used to highlight advanced problem-solving strategies, such as optimizing for time and space complexity, following systematic coding procedures, and leveraging computational tools for mathematical reasoning.

**Blue** emphasizes critical thinking and analytical approaches, including problem simplification, decomposition, assumption identification, and the formulation of step-by-step solution plans.

**Teal** indicates the need to flexibly adapt the provided thoughts and strategies to the specific context of the current task, ensuring relevance and effectiveness.

**Instructor Agent:**

**Prompt for Advancing the Current Task**

<NOTICE>
Provide me with the next instruction.
Before producing the final answer, please check whether I have rechecked the final answer using different toolkit as much as possible. If not, please remind me to do that.
My answer's confidence is {confidence:.2f}, the range is [0,1]
If you find that my answer has been repeatedly modified or is inconsistent, please confirm it extra carefully, do not accept my new answer directly.
Please use multiple tools for verification and check whether I have fallen into a mental trap.
If I am currently exploring a new approach, please remind me to continue exploring and do not return to the old approach directly.
**[IMPORTANT NOTICE]**

- If confidence is low or I have rethink in this round, be cautious, confirm my solution, and do not output <CURRENT_TASK_OVER> or move forward to the next step.
- If the rethink identifies fundamental flaws, immediately switch methods or tools to avoid getting stuck.
- You can give me appropriate instructions based on the rethink. This is the only way to prevent AI from falling into infinite loops or path dependencies.
- Only when the rethink has not rejected the current approach, we can continue.

</NOTICE>

Here is the current task: <task>{current_task}</task>, we should do this task step by step.
The next task is: <next_task>{self._get_next_task()}</next_task>
You should think about why we should do the current task, what does it mean to the overall task, and what is the relationship between the current task and the next task.
Based on your think result, you should give me the next instruction.
The task may be very complicated, so you should give me the instruction step by step.

**Notice:**

**Blue** is used to emphasize critical verification steps, such as cross-checking answers with multiple toolkits, being cautious with low-confidence or inconsistent answers, and actively preventing logical traps or premature acceptance of solutions.

**Violet** highlights the importance of methodological innovation and reflection, including exploring new approaches, switching strategies when fundamental flaws are detected, and following rethink-driven instructions to avoid repetitive or path-dependent errors.

**Magenta** draws attention to task-level meta-cognition, prompting consideration of the purpose and significance of the current task.

**Prompt for Dynamic Task Simplification**

To better proceed with the following process, please provide a comprehensive summary based on the historical messages, overall task, and current task.

**Context Information:**

- Overall task: <task>{self.original_task_prompt}</task>
- The subtask to do: <current_task>{current_task}</current_task>

**Please structure your summary in the following format:**

1. <completed_conversations>

    (1) Only summarize instructions and solutions that have actually contributed to progress, such as major strategy shifts, key conclusions, critical failures, or substantial changes after rethink.

    (2) Ignore repetitive, template-based, or ineffective content.

    (3) Do not guess or assume any information; only summarize based on historical messages.

    </completed_conversations>

2. <optimized_problem>

    (1) Reframe the current task in concise language, incorporating previous successes and converting solved sub-problems into known conditions.

    (2) Clearly identify which ideas or approaches have been excluded and which conditions have been proven.

    (3) Make the problem statement self-contained, only retaining key conditions and objectives, and avoid including irrelevant historical details.

    (4) Do not perform any mathematical calculation or formula derivation.

    </optimized_problem>

3. <global_obstacles>

    (1) Summarize the main types of obstacles, deadlocks, or common pitfalls encountered, merging similar issues to avoid repetition.

    (2) Point out which ideas or methods have failed multiple times and which problems remain unsolved.

    (3) If possible, briefly analyze the reasons for these obstacles (such as misunderstanding, implementation details, or limited thinking).

    </global_obstacles>

4. <next_strategies>

    (1) Based on the above global obstacles, summarize the directions that have not been fully explored or validated in history, or point out which ideas can be further refined.

    (2) Propose 1-2 concrete suggestions for improvement, which may include: changing the mathematical modeling, reconstructing equations, etc.

    </next_strategies>

**Finally, output your complete summary in the format:**
<summary>[Your structured summary following the above format]</summary>

**Actor Agent:**

**Enabling Long-term Memory**

- Use the `recall_experience` tool to get experience or review history.

- If you encounter repeated failures, or suspect the problem is a classic or well-studied one, use search tools to look up standard results or relevant literature.

### General Principles

- Stay rigorous, always check the accuracy of your solution!

- Always use tools for mathematical calculations and pattern derivation; do not calculate by yourself. For each conclusion, use tools to verify it—correct verification is more important than incorrect answers.

- If you find that your results are inconsistent, analyze the reasons for the inconsistency and verify further.

- Never rely solely on extrapolation or pattern fitting; always provide a mathematical justification or proof for your approach.

- Don't use too many tools in one step. After successfully using a tool, you can stop and summarize the result; I will adjust the instruction based on the result.

### Validation and Testing

- After obtaining a result, always cross-validate it using a different method or tool (e.g., sympy, code, or manual calculation for small cases).

- Whenever you derive a new formula or approach, first test it on small-scale examples (e.g., $n = 3, 4, 5$) and compare with direct enumeration to ensure correctness before generalizing.

- Before large-scale reasoning, first use small-scale examples for strict testing, and continuously optimize the method during the testing process.

- If you encounter an unexpectedly large number in your calculation or output, pause and re-examine your approach; this often signals a fundamental flaw in the method or an invalid use of brute-force enumeration.

### Mathematical and Combinatorial Problems

- When solving combinatorial or set-counting problems, avoid brute-force enumeration over all subsets if the set is large. Instead, consider using mathematical methods such as the inclusion-exclusion principle, combinatorial analysis, or generating functions to count the number of valid subsets efficiently.

- If a mathematical method (such as inclusion-exclusion, combinatorial formula, or symbolic manipulation) can be implemented using code or sympy, always try to use these tools for verification and calculation, rather than relying on manual reasoning or intuition.

- For problems involving divisors, LCM, GCD, or subset properties, try to express the problem in terms of prime exponents and use combinatorial logic to count valid cases. Use code or sympy to automate the process whenever possible.

### Probability and Counting Problems

- For probability/counting problems, avoid direct derivation. Use code for recursive counting to ensure no solutions are missed. Probability problems must be solved with code, and the code must be thoroughly checked to ensure all requirements are met and all cases are covered. If combinatorial formulas are used, they must be cross-validated with code and corrected until accurate. Alternatively, use code to directly output the answer.

- For probability problems, validation with small-scale examples is mandatory. If the current answer has not been cross-validated, immediately output <ERROR> and indicate that validation is needed.

- For probability problems, first use search tools to identify a reliable solution and fully understand it. Direct derivation without this step is likely to lead to incorrect conclusions.

### Geometry Problems

- If this is a geometry problem, do not rely on plotting (you do not have the ability to view images). Instead, try to use logical analysis methods.

**Rethink Agent:**

**You are a justice judge who judges whether the solution is consistent with the current task.**

Please review the history messages. The current solution may be in a wrong way to the original <task>{self.task_prompt}</task>. Please check if the current method is too complex, and whether the current answer has considered all the conditions in the question. You need to check whether the user and assistant are solving the problem by using code or sympy or other tools.

**Special attention: You must actively check if the current approach is stuck in 'mental set' or 'global misconceptions'. If you find repeated circling around a detail, please prioritize considering:**

- Can we switch to a completely different modeling approach?
- Can we use more concise approaches like global constraints or distribution equations?
- Can we try minimal special cases or counterexamples?
- Do we need to actively consult external materials or classic solutions?

Please make sure to provide specific suggestions on [how to break out of the original thinking] (with how to break out of the original thinking), not just fixing details.

Based on above, if you find anything, wrap your findings with <advice> and </advice>, for example:
<advice>
error:
reason:
suggestion: summarize the main global obstacles or deadlocks, and propose at least one concrete suggestion for breaking the deadlock (such as changing the modeling method, trying small-scale examples, or consulting external resources).
</advice>

If you have no advice, add <NO_RETHINK> at the beginning of your answer, do not add any other content.

**Notice:**

**Violet** is used to highlight the identification of over-complicated reasoning, mental set, global misconceptions, and the need for critical evaluation of the current approach.

**Magenta** is used to emphasize actionable suggestions for breaking out of original thinking patterns, focusing on proposing concrete strategies to overcome major obstacles or deadlocks.

**Confidence Evaluator**

The Confidence Evaluator is designed to assign a confidence score to the textual outputs generated by large language models, with the score ranging from 0.0 to 1.0. This method innovatively integrates two primary factors: the presence of uncertainty-related keywords and the entropy distribution of the text.

First, the evaluator scans the generated text for typical uncertainty keywords, such as "wait", "hold", "but", "okay", "hmm", "not_sure", "error", "mistake", "inconsistent", "incorrect", "wrong", as well as the phrase "not sure". If any of these terms are detected, the confidence score is penalized accordingly to reflect the model's uncertainty in its response.

Second, the evaluator computes the character-level entropy of the text, which serves as a measure of the diversity and unpredictability within the content. Higher entropy values generally indicate greater uncertainty in the generated output.

Finally, the evaluator combines the effects of these two components through a weighted integration, resulting in a normalized confidence score. Based on this confidence score, a rethink process may be triggered if the score falls below a certain threshold. This information is then communicated to the

Instructor Agent, enabling dynamic adjustment of the agent's level of caution and decision-making strategy.

## C.2 MEMORY SYSTEM

Our memory system is composed of two distinct modules: **working memory** and **long-term memory**, each designed to enhance the agent's reasoning and information management capabilities.

**Working Memory Implementation:**

The `work_memory_limit` parameter constrains the capacity of the working memory, specifying the maximum number of recent messages that the agent can directly access during the current task. Before any historical message is written into working memory, it is preprocessed by the `_simplify_content` function. This function removes redundant or irrelevant information, such as specific tags (e.g., <PASS>), system prompts, failure notifications, and unrelated auxiliary explanations, thereby retaining only the core content that is valuable for subsequent reasoning and memory management.

Messages are transferred from working memory to long-term memory under two circumstances: (1) when a state transition occurs, or (2) when the working memory exceeds the `work_memory_limit`. During this transfer, a series of semantic and structural preprocessing steps are performed, with differentiated handling based on message type. For example, messages containing historical markers, tool invocation results, instructions, and solutions are embedded and organized using tailored strategies. Tool invocation results are filtered to exclude invalid or error-containing outputs, retaining only execution results with practical reference value, and are truncated as needed for efficient retrieval. For instruction-solution pairs, the latest instruction and its corresponding solution are concatenated to form a composite memory unit with enhanced contextual relevance. All content to be stored is converted into vector representations via an embedding model prior to storage.

**Long-term Memory Implementation:**

We introduce a dedicated tool, `LongTermMemoryToolkit`, accessible to both the Instructor Agent and Actor Agent. This tool enables agents to semantically retrieve relevant long-term memories from a vector database based on provided keywords, as well as to search an experience pool for related experiential knowledge. Both agents autonomously determine when to invoke the tool and which keywords to use for retrieval, thereby supporting flexible and context-aware memory utilization.

This dual-memory architecture, combined with advanced content simplification and semantic retrieval mechanisms, enables our framework to dynamically manage information, prevent memory overload, and facilitate efficient long-term knowledge integration—demonstrating a novel and biologically inspired approach to memory management in multi-agent systems.

## C.3 CASE STUDY

We selected a representative problem from the ablation experiments in Section B.7 and compared the execution logs of our framework and the CoT method on this problem. This problem comes from MATH_LV5 (see Section B.5).

Figure 4 shows the execution log for our framework, with each box representing a distinct system state. The content within each box corresponds to the working memory at that stage. For example, at the end of the leftmost column, the Instructor Agent outputs <CURRENT_TASK_OVER>, which triggers state simplification: all relevant information in the current working memory is transferred to long-term memory. When transitioning to the next state (the second column from the left), the working memory is reset to contain only the current subtask and a summary. Based on these, the Instructor Agent and Actor Agent continue their dialogue.

In the fourth column, the system successfully retrieves code from long-term memory using the `recall_experience` tool, and this retrieved code is then used for subsequent calculations. This demonstrates the effectiveness of integrating both working and long-term memory.

It is important to note that not every <CURRENT_TASK_OVER> output triggers a state transition. State transitions occur only when the number of interactions reaches a certain threshold. For instance, transitions between the second and third columns are due to reaching this threshold. In the third

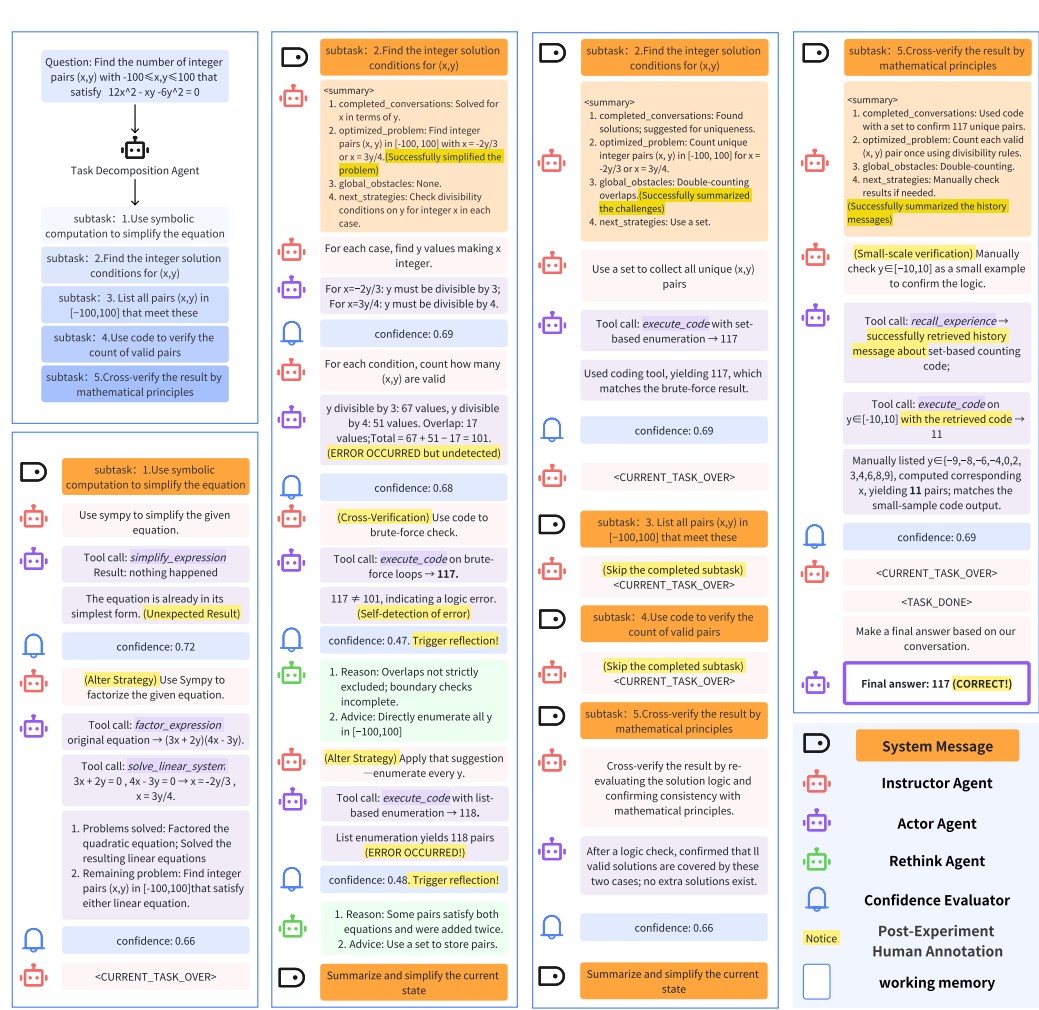

Figure 4: Execution trace of our framework on a MATH_LV5 problem, illustrating task decomposition and dynamic simplification, error correction, and long-term memory retrieval. **Highlighted segments indicate human annotation for reference.**

column, although several subtasks are completed, no state transition occurs because these subtasks are quickly skipped; performing a transition and summary in such cases would consume excessive computational resources.

Throughout the figure, we highlight key information, such as occurrences of unexpected results and strategy alterations. These annotations illustrate the robustness of our system in adapting to special cases and handling errors.

To provide a comprehensive comparison, the execution log for the CoT method on the same problem is presented below:

---

**Question:** Find the number of integer pairs $(x, y)$ with $-100 \leq x, y \leq 100$ that satisfy $12x^2 - xy - 6y^2 = 0$

**Step 1:** Understand the structure of the equation

Treat $12x^2 - xy - 6y^2 = 0$ as a quadratic in $x$.

**Step 2:** Solve for $x$ using the quadratic formula

Apply the formula to obtain $x = \frac{3y}{4}$ or $x = -\frac{2y}{3}$.

**Step 3:** Identify all integer solutions

For $x = \frac{3y}{4}$, $y$ must be a multiple of 4; for $x = -\frac{2y}{3}$, $y$ must be a multiple of 3. This leads to two sets of integer pairs: $(3k, 4k)$ and $(-2m, 3m)$, where $k$ and $m$ are integers such that $x, y \in [-100, 100]$. Each set contains 67 pairs.

**Step 4:** (Key Error Step) Check for overlap between the two sets

At this step, the solution simply checked whether $(0, 0)$ appeared in both sets and assumed it was the only overlapping pair. However, this overlooks all pairs where $y$ is a common multiple of 3 and 4 (i.e., $y$ is a multiple of 12), which actually results in multiple overlapping pairs.

**Step 5:** (Key Error Step) Compute the total number of distinct ordered pairs

Based on the previous step, the solution added the sizes of both sets and subtracted one for $(0, 0)$, obtaining 133. This method fails to account for all overlapping pairs, leading to an over count. The correct approach is to subtract the total number of pairs where $y$ is a multiple of 12, resulting in the correct answer of 117.

**Summary:**

Steps 1-3 are correct.

The key issue in Steps 4-5 is that the error in counting overlapping pairs was neither detected nor corrected. This oversight resulted in an overestimate of the final answer. **Effective error detection and correction would require a thorough inclusion-exclusion analysis for all overlapping pairs**, rather than assuming only $(0, 0)$ as the overlap.

---

**Notice:** Purple is used to emphasize the nature and reasoning of the error.

These two cases demonstrate that our approach is capable of both error correction and memory retrospection, which are crucial for solving complex mathematical problems.

# D  COMPUTATIONAL RESOURCE ANALYSIS

The number of reasoning rounds is adaptively determined based on task complexity. For simple tasks, the framework typically requires 5 rounds, consuming approximately 30k tokens, whereas complex tasks may require up to 20 rounds, with a total token usage ranging from 100k to 150k.

For example, coding tasks are generally less complex, with most problems requiring between 30k and 50k tokens. MATH tasks are moderately more challenging, typically consuming 50k to 80k tokens. In contrast, AIME tasks are the most demanding, often requiring more than 100k tokens to complete.

**Notably, although our framework introduces additional mechanisms such as periodic summarization and reflection, the overall token consumption remains efficient due to several design choices:**

- Reflection is triggered either based on confidence thresholds or proactively by the Actor Agent, rather than at every step, thus avoiding unnecessary overhead.

- We implement working memory pruning to prevent the context from becoming excessively long.
- Messages irrelevant to the current task are moved to long-term memory, further reducing the active context size.

As a result, the average number of tokens required per reasoning round is less than or equal to that of the original CAMEL framework, upon which our system is built.

# E    GUIDELINES FOR ADAPTING AND OPTIMIZING THE FRAMEWORK FOR NEW TASKS

Our framework exhibits strong generalization capabilities. As demonstrated by our experiments on code-related tasks, it can be directly applied to a variety of tasks without any modifications, consistently achieving robust performance. Furthermore, if adaptation is required, the framework is highly flexible and easy to modify. Even minor adjustments can lead to substantial performance improvements, as evidenced by our results on the MATH dataset, particularly in the counting and probability category.

Based on our experience, we recommend the following strategies to maximize performance when applying the framework to new domains or problem types:

**1. Adjusting the Reflection Threshold:** Carefully tune the confidence threshold that triggers the reflection (rethink) mechanism according to the complexity and uncertainty of the target task. Increasing this threshold can prompt the agent to engage in self-reflection and correction more frequently when faced with ambiguous or low-confidence reasoning results. This helps to reduce error accumulation and enhances the robustness and accuracy of the reasoning process.

**2. Integrating Domain-Specific Experience into Prompts:** Systematically review and incorporate domain-specific knowledge, common strategies, and effective heuristics into the prompt design. By explicitly emphasizing typical pitfalls, error-prone scenarios, and efficient solution approaches within the prompts, users can significantly improve the model's generalization and adaptability to new tasks.

**3. Leveraging More Advanced Base Models:** Consider upgrading to more powerful language models as they become available. For example, models such as GPT-4.1 Mini exhibit stronger instruction-following capabilities, more sophisticated reasoning, and better task comprehension compared to earlier versions. In practice, these models can more accurately interpret multi-step instructions and adhere to complex workflows, leading to higher task completion rates and improved answer quality. Sometimes, simply upgrading the base model for key agents such as the Rethink Agent or Actor Agent can also yield noticeable performance improvements.

By following these guidelines, users can effectively tailor and optimize our framework for a variety of novel tasks and application scenarios.

# F    FUTURE DIRECTIONS FOR BRAIN-INSPIRED MULTI-AGENT SYSTEM OPTIMIZATION

## F.1    CONTINUAL LEARNING AND SELF-EVOLUTION IN MODULAR AGENTS

Inspired by the brain's lifelong learning and adaptability, each module in our framework can be enhanced for continual self-improvement through two main approaches:

First, each module can implement a data flywheel by continuously collecting decision data and feedback, and periodically refining its models or strategies. When retraining is not feasible, a structured experience pool can be maintained to record both successes and failures. Through ongoing self-reflection and consolidation of experiences, modules can adapt and optimize over time, analogous to the brain's processes of experience summarization and memory consolidation.

Second, modules can dynamically adjust their internal rules or parameters during operation, rather than relying on static execution. For example, modules can track meta-information such as confidence, recent success or failure, and citation frequency for each memory, enabling adaptive retrieval

strategies that prioritize high-confidence and recently successful long-term memories. Reflection and verification mechanisms can be further enhanced by introducing meta-cognitive modules that assess the quality and specificity of suggestions, dynamically adjust confidence thresholds based on context and task progress, and monitor the reliability of tool invocation and validation. Additionally, meta-monitoring agents can proactively detect signs of stagnation or deadlock and trigger strategy switching, while frustration signals inspired by the amygdala and dopaminergic systems can prompt adaptive exploration or external assistance when repeated failures occur.

### F.2 Enhancing Accuracy through Additional Brain-Inspired Modules

To further enhance accuracy, additional modules inspired by specific brain mechanisms can be incorporated, each grounded in established neuroscience.

**1. Hierarchical Meta-Control Modules**
Inspired by the dorsolateral and anterior prefrontal cortex (dlPFC, aPFC), a meta-controller can periodically adjust task decomposition strategies and granularity, dynamically merging or splitting subtasks for optimal performance.

**2. Conflict Monitoring and Error Detection**
Drawing from the anterior cingulate cortex (ACC), a dedicated agent can monitor for logical inconsistencies and conflicts, issuing alerts and prompting plan revisions when necessary.

**3. Counterfactual Reasoning and Hypothesis Testing**
Mirroring the brain's capacity for alternative scenario evaluation, mechanisms for counterfactual reasoning and hypothesis testing can be added to validate key conclusions and prevent error propagation.

**4. Memory Consolidation and Replay**
Inspired by hippocampal replay, the system can periodically review and verify critical long-term memories, correcting earlier conclusions when anomalies are detected.

### F.3 Enhancing Domain-Specific Capabilities through Modular Extension

To further improve the framework's adaptability and performance in specific or real-world scenarios, our architecture supports the integration of specialized agents tailored to particular tasks. For example, in retrieval-augmented generation (RAG) tasks, a dedicated knowledge retrieval and integration agent can be introduced; in online shopping scenarios, a contextual awareness agent can dynamically aggregate user history, environmental information, and real-time interactions to infer user intent and preferences; and in multimodal applications, a multimodal perception agent can fuse information from text, images, and audio sources.

Although our workflow is highly interdependent, the system supports modular extension: the core layer handles main reasoning, while extension agents are flexibly integrated as plug-ins and activated by the Instructor Agent for specific scenarios. The Instructor Agent can manage global and knowledge-oriented modules, while the Actor Agent can execute concrete tasks and report results. This separation of responsibilities, together with modular extensibility, can enable efficient adaptation to diverse real-world applications.

