# OpenReview forum: "BMAS: A Brain-Inspired Multi-Agent System with PFC-Guided Task Coordination and Hippocampus-Neocortex Dual Memory for Scalable Multi-Step Reasoning"
_ICLR.cc/2026/Conference — Submitted to ICLR 2026_

### Official Review · Reviewer_1sGq · 2025-10-23

**Soundness:** 2
**Presentation:** 2
**Contribution:** 2
**Rating:** 2
**Confidence:** 4

**Summary:**

This paper identifies several shortcomings of current multi-agent systems and draws inspiration from the human brain to propose a Brain-Inspired Multi-Agent System, where different modules of the multi-agent framework correspond to distinct regions of the brain. Designing agent architectures based on brain mechanisms is a meaningful direction, and it may enable principled planning and coordination pathways, as well as efficient memory structures as the authors suggested.

I reviewed this paper before during the NeurIPS 2025 review process. Unfortunately, it seems that the valuable comments raised by the reviewers at that time have not been addressed or incorporated into this version.

**Strengths:**

Using the analogy of the human brain to understand multi-agent systems is an interesting and insightful perspective. The ablation study in Section 4.4, which examines the relationship between the memory mechanism and turn and accuracy, is well-designed.

**Weaknesses:**

I have compared the previous version with the current one. Unfortunately, some important weaknesses from the last reviews remain unresolved:

1. Unsupported claim on efficiency [Line 32]. There is no comparison or experiments using any efficiency metrics, so the claim remains unsubstantiated.

2. Limited scope of experiments and weak baselines. Although the authors propose a multi-agent framework, no agentic tasks are evaluated.

3. Limited novelty and lack of a solid theoretical foundation for the advantages of the proposed MAS.

**Questions:**

Could the authors provide details on what changes or improvements have been made in this version compared to the previous one?

---

### Official Review · Reviewer_7BVJ · 2025-10-26

**Soundness:** 3
**Presentation:** 1
**Contribution:** 3
**Rating:** 4
**Confidence:** 3

**Summary:**

This paper proposes BMAS, a novel brain-inspired framework for multi-agent systems, designed to overcome the limitations of static architectures and inefficient memory management. The system introduces a module that dynamically decomposes tasks and coordinates agents, moving beyond rigid, predefined structures. This is complemented by a dual-memory system which selectively consolidates and retrieves relevant information to mitigate context inflation in long-horizon tasks. The authors validate BMAS on complex mathematics and coding datasets, demonstrating improved efficiency, stability, and reasoning robustness. This approach effectively unifies architectural design and memory management, offering a more scalable solution for multi-step reasoning.

**Strengths:**

1.  **Novel Task Management:** The design featuring **hierarchical task decomposition and dynamic simplification** is an intelligent approach to managing complex problems, improving adaptability over static methods.

2.  **Strong Empirical Performance:** The proposed system **consistently outperforms strong MAS baselines** across diverse and challenging tasks, demonstrating its practical effectiveness and robustness.

**Weaknesses:**

1.  **Poor Presentation:** The paper's clarity suffers from a cluttered presentation. Figure 1 and Section 3.2 are crowded with biological analogies, lacking a clean, abstract illustration of the architecture. The main process diagram is cognitively overloaded, while the task decomposition figure is overly simple, failing to effectively demonstrate the proposed hierarchical simplification.

2.  **Limited Related Work:** The related work section is insufficient. Section 2.1 uses only a few citations to represent entire research areas and does not discuss the experimental baselines. Section 2.2 strangely includes "Least-to-Most" prompting and "Atom-of-Thought," which are not memory-oriented methods, without properly introducing the MAS systems that supposedly use them.

3.  **Inconsistent Evaluation Protocol:** The evaluation protocol is inconsistent, using different LLMs across experiments (Qwen3-Plus in Table 1, GPT-4-Turbo/mini in Table 2, GPT-4.1 mini in Table 3). This raises questions about whether the system's effectiveness is dependent on specific model implementations for different tasks.

4.  **Insufficient Ablation Study:** The ablation study in Section 4.4 is inconclusive. By removing both the task management and memory modules at the same time, it fails to independently explore the contribution of each component, making it impossible to assess their individual impact.

**Questions:**

See above

---

### Official Review · Reviewer_qeZR · 2025-10-29

**Soundness:** 1
**Presentation:** 2
**Contribution:** 1
**Rating:** 2
**Confidence:** 4

**Summary:**

This paper proposes a BMAS multi-agent system comprised of modules inspired by how the human brain solves complex tasks through decomposition, planning and working and long-term memory. The authors claim that BMAS addresses the adaptive coordination challenge in MAS and applies the hierarchical task simplification to avoid static architectural choices. The framework is evaluated on AIME, MATH, and HumanEval datasets, showing improvements over the single model baseline and some existing multi-agent frameworks.

**Strengths:**

- Brain-inspired analogies in system workflow and agent modules provide an elegant motivation for the architecture design choices.
- The paper evaluates the system across two domains (mathematics and coding) using three different datasets (AIME, MATH, HumanEval).
-  The system shows measurable improvements over baselines, particularly on the challenging AIME dataset.

**Weaknesses:**

1. The paper claims "dynamic coordination" as a key contribution, but the actual mechanism remains unclear. The agent interaction sequence looks fixed (Task Decomposition -> Instructor -> Actor -> Confidence Evaluator -> Rethink -> Instructor).
2. The paper mentions "hierarchical task simplification" repeatedly but does not clearly explain what makes it hierarchical. The task decomposition appears to be a flat list as explicitly stated in the Task Decomposition Agent prompt in Appendix C.1: "3-5 subtasks total, each subtask should not have sub-tasks").
3. The paper claims superiority over systems with central schedulers, but the Instructor Agent appears to function as a central coordinator, controlling task flow and issuing instructions.
4. OWL is chosen as a baseline for AIME based on GAIA benchmark performance, which is not relevant to mathematical reasoning tasks. The authors should compare against multi-agent architectures tested on mathematical reasoning in general or on AIME in particular.
5. AutoGen is listed as a baseline but AutoGen is a framework, not a fixed architecture. The exact multi-agent system architecture built with AutoGen is not specified, making the comparison meaningless.
6. No analysis is provided on how performance varies with different backbone models. For a framework claimed to be a general-purpose multi-agent solution, it should demonstrate consistent improvements across multiple LLM backbones.
7. Claims about the major performance gain because of multi-stage reasoning, dynamic memory management, and reflection mechanisms in Sections 4.2 and 4.3 are not supported with any explicit evidence or analysis.
8. The paper claims (Section 4.2) BMAS is a "general-purpose, requiring no task-specific role configuration", but: (i) the Task Decomposition Agent prompt (Appendix C.1) contains explicit instructions for coding and math problems; (ii) the Actor Agent prompt (Appendix C.1) provides detailed instructions for specific math problem types; (iii) in the guidelines for adaptation (Appendix E) it is recommended to "integrate domain-specific experience into prompts" contradicting the general-purpose claim.

**Questions:**

1. Where exactly does dynamic coordination occur if the pipeline is predetermined? Can the system break this fixed sequence based on runtime needs?
2. How is the hierarchical task decomposition implemented?

---

### Official Review · Reviewer_1tpk · 2025-11-01

**Soundness:** 2
**Presentation:** 2
**Contribution:** 2
**Rating:** 4
**Confidence:** 3

**Summary:**

The authors propose a brain-inspired multi-agent system (BMAS), consisting of a prefrontal cortex inspired module for hierarchical task decomposition and dynamic adjustment of tasks based on feedback, and a dual memory system, consisting of working memory and long term memory. Specifically BMAS consists of five modules dynamically interacting with each other - Task Decomposition Agent, Instructor Agent, Actor Agent, Confidence Evaluator, and Rethink Agent—together with a two-tier memory architecture inspired from hippocampus-neocortex. Working memory contains information relevant for the current subtask, and long term memory selectively stores filtered working-memory traces and supports semantic retrieval by both the Instructor and Actor agents. BMAS shows improvements on mathematical and coding tasks.

**Strengths:**

1. A brain inspired multi agent system is proposed for complex multistep reasoning tasks, consisting of modules like task decomposition, reflection, dual memory system containing task specific information, storing historical contexts and retrieval for long term reasoning
2. Improvements on mathematical and coding tasks

**Weaknesses:**

1. Recent work proposed modular agentic planner (MAP) [1], consisting of different modules like task decomposer, monitor, actor, evaluator etc inspired from different brain regions. The authors don’t discuss how BMAS differs from MAP and should also be evaluated as a baseline for comparison.

2. The different agents requires lots of specific instructions and prompts, its not clear how sensitive is BMAS to prompt engineering, and also how would this generalize to complex multi-step reasoning/planning tasks, except mathematics and coding.

3. There are no analysis of failure modes, specifcially which modules are primarily bottlenecks.

[1] - Webb, T., Mondal, S.S. & Momennejad, I. A brain-inspired agentic architecture to improve planning with LLMs. Nat Commun 16, 8633 (2025). https://doi.org/10.1038/s41467-025-63804-5

**Questions:**

1. Some targeted ablations would be helpful in understanding which mechanisms are most important for BMAS agent, like how does it perform just without the task decomposer, just without long-term memory, just without reflection etc?


2. How does the performance vary with confidence threshold?


3. Unlike parallel approaches
such as ”Tree of Thought” (Yao et al., 2023), humans reason along a single path, using self-correction
and strategy shifts for creativity. - humans reason along single path doesn’t seem correct, can the authors cite evidence for that?

---

### Meta-Review · Area_Chair_kdsS · 2026-01-06

**Summary:**

Reviewers consistently express concerns regarding mechanistic clarity, substantiation of claims, and experimental validity. While the brain-inspired framing is generally viewed as a reasonable motivation, multiple reviewers argue that key claimed contributions—particularly dynamic coordination and hierarchical task decomposition—are not convincingly supported by the described system. The agent interaction pipeline appears largely predetermined, with limited evidence that coordination or task structure adapts meaningfully at runtime.

Several reviewers also question the novelty and positioning of the work relative to prior brain-inspired or modular multi-agent systems, noting insufficient comparison to closely related architectures and unclear differentiation. On the empirical side, although improvements are reported on math and coding benchmarks, reviewers raise concerns about baseline choice, evaluation protocol consistency, lack of targeted ablations, and limited analysis of failure modes, which collectively weaken the evidential support for the paper’s claims. Finally, the lack of author engagement during the discussion phase leaves these concerns unresolved.

**Reviewer Concerns:**

No rebuttal was provided by the authors, and therefore none of the substantive reviewer concerns were addressed during the discussion phase.

**Reviewer Scores:**

see above

---

### Decision · Program_Chairs · 2026-01-26

Reject